# Progression and Trends in Virus from Influenza A to COVID-19: An Overview of Recent Studies

**DOI:** 10.3390/v13061145

**Published:** 2021-06-15

**Authors:** Hakimeh Baghaei Daemi, Muhammad Fakhar-e-Alam Kulyar, Xinlin He, Chengfei Li, Morteza Karimpour, Xiaomei Sun, Zhong Zou, Meilin Jin

**Affiliations:** 1State Key Laboratory of Agricultural Microbiology, Huazhong Agricultural University, Wuhan 430070, China; sanaz.baghaei@yahoo.com (H.B.D.); cfli@webmail.hzau.edu.cn (C.L.); sunxm320@126.com (X.S.); 2College of Veterinary Medicine, Huazhong Agricultural University, Wuhan 430070, China; fakharealam786@hotmail.com; 3Key Laboratory of Development of Veterinary Diagnostic Products, Ministry of Agriculture, Wuhan 430070, China; 4Department of Biology, Azad University of Rasht, Rasht 4147654919, Iran; morteza_zmmm@yahoo.com

**Keywords:** COVID-19, influenza A, coronavirus, pandemic

## Abstract

Influenza is a highly known contagious viral infection that has been responsible for the death of many people in history with pandemics. These pandemics have been occurring every 10 to 30 years in the last century. The most recent global pandemic prior to COVID-19 was the 2009 influenza A (H1N1) pandemic. A decade ago, the H1N1 virus caused 12,500 deaths in just 19 months globally. Now, again, the world has been challenged with another pandemic. Since December 2019, the first case of a novel coronavirus (COVID-19) infection was detected in Wuhan. This infection has risen rapidly throughout the world; even the World Health Organization (WHO) announced COVID-19 as a worldwide emergency to ensure human health and public safety. This review article aims to discuss important issues relating to COVID-19, including clinical, epidemiological, and pathological features of COVID-19 and recent progress in diagnosis and treatment approaches for the COVID-19 infection. We also highlight key similarities and differences between COVID-19 and influenza A to ensure the theoretical and practical details of COVID-19.

## 1. Introduction

The coronavirus infection (COVID-19) has recently emerged as a serious concern at the World Health Organization (WHO). It is a widespread outbreak in humans with a high mortality rate. The outbreak of this virus commenced in China in December 2019 and then spread rapidly to many other countries in a short time. Coronaviruses are often known to mutate, recombine and be conducive to challenge the diagnosis and treatment process.

It should be noted that the risk of a coronavirus outbreak depends on its features, including its prevalence rate among individuals, the severity of disease, reliable medical diagnosis tools, and vaccines or drugs available to control the dangerous effects of the virus [1]. Conspiracy theories regarding the outbreak’s causes have flourished as policymakers and research institutions continue investigating what caused it. As a result, it is important to pinpoint the virus’s exact roots as well as the vectors through which it spreads. From a public health perspective, it is also crucial to consider the impact that conspiracy theories regarding COVID-19’s roots can have on the public’s perceptions of the virus’s health and policy ramifications, and their ability to participate in prosocial activities to slow its dissemination [2]. At the start of the pandemic, it was suggested that the origin of COVID-19 was from bats; however, owing to a lack of phylogenetic network, it is unclear if bats directly move to humans or transmit to intermediate hosts to allow animal-to-human transmission [3]. The reason behind this is the continuous change in viral sequences. As a result, constructing a phylogenetic network is critical for investigating the virus’s adaptation and determining its origin [4].

In the twenty-first century, the first influenza A pandemic occurred in Mexico in early March 2009. Then, some similar cases appeared in the US and many other countries. In the first days, the disease was known as the swine flu pandemic, but a short time later, the WHO used the term Influenza A (H1N1) [5,6,7]. Influenza is a respiratory infection that can infect 5–15% of the human population annually. Via antigenic drift, influenza A virus can result in new influenza A subtypes with pandemic potential, such as influenza A/H1N1 subtypes that show more severe clinical symptoms than the other subtypes [8,9,10].

Furthermore, COVID-19 and influenza viruses have very similar signs, and symptoms may explain the similar origin [11]. According to a recent World Health Organization survey, the COVID-19 attack and disease burden in children have been much lower than influenza outbreaks, and the secondary household attack rate has also been low. This is in stark contrast to reports of the virus spreading quickly in enclosed spaces like hospitals or cruise ships, as well as a high prevalence of healthcare-associated infections [12].

This narrative analysis report aims to summarize and compare early studies on the epidemiology, clinical effects, diagnosis, treatment, and prevention of COVID-19 and influenza A to provide theoretical and practical details.

## 2. Virology and Structural Characteristics

### 2.1. Coronavirus (COVID-19)

Coronavirus belongs to the family of Coronaviridae. Coronaviridae is a family of enveloped, positive-strand RNA viruses that infect amphibians, birds, and mammals [13]. The virus’s genome size is between 26 to 32 kB, which is considered among the viruses with the most extensive RNA. These viruses have two different surface proteins that are named after their external features [14]. Coronaviridae family is divided into two subfamilies, including Coronavirinae and Torovirinae. The Coronavirinae subfamily is composed of four well-known genera (i.e., alpha coronavirus (α-coronavirus), beta coronavirus (β-coronavirus), gamma coronavirus (γ-coronavirus,) and delta coronavirus (δ-coronavirus)). Human diseases are associated with the genera alpha coronavirus and beta coronavirus, while those belonging to the genera gamma coronavirus and delta coronavirus cause disease in animals. COVID-19 as SARS-CoV (Severe Acute Respiratory Syndrome) and MERS-CoV (Middle East Respiratory Syndrome) belong to the beta coronavirus genus, while HCoV-NL63 and HCoV-229E belong to the alpha coronavirus genus [13]. Coronavirus can infect mammals and birds by triggering various lethal diseases [15].

COVID-19 is the seventh coronavirus known to infect humans, and its clinical symptoms are similar to influenza A, ranging from asymptomatic or mild through severe disease and death. The COVID-19 genome is 50% and 79%, the same as MERS-CoV and SARS-CoV, respectively. The Coronaviruses belong to the enveloped, non-segmented, single-stranded RNA, positive-sense viruses whose genome size is 26–32 kB (nearly 30 kB). Two important ORFs were found in the COVID-19 genome. They are ORF1a and ORF1b, with the potential to encode 16 non-structural proteins (NSP), constituting almost two-thirds of the viral genome.

Moreover, the remaining one-third of the COVID-19 genome encodes structural and accessory proteins. Diverse structural proteins including S (spike), M (membrane), N (nucleocapsid), and E (envelope) were identified in the COVID-19 virion. These structural proteins are critical mediators in cell attachment and entrance, genome replication, and pathogenicity and finally promote infection [15].

The highly glycosylated COVID-19 S protein comprises two subunits called S1 and S2 (Figure 1). S1 with a receptor-binding domain (RBD) plays a crucial role in cell entrance and tissue tropism by binding to ACE2 on the host cell. Furthermore, a polybasic cleavage site (PRRAR) recognized by the prototype proprotein convertase furin was identified at S protein. This pre-processing gives an unusual feature to COVID-19 that significantly increases tissue tropism and transmissibility [15].

Moreover, TMPRSS2 is a transmembrane serine protease that facilitates cleavage at the S1-S2 boundary and downstream cell fusion. N protein appears to be directly bound to the RNA genome and involved in ribonucleoprotein construction, which functions in cell fusion and entrance. Furthermore, N protein functioning as a cell cycle inhibitor was determined by cell cycle studies. M protein is involved in ribonucleoproteins formation and is considered a key mediator of inflammatory responses in the host, whereas virion formation and viral pathogenicity are triggered by E protein [16]. COVID-19 genomic RNA has also been demonstrated to contain a 5’-cap structure and a 3’-polyadenylation tail, essential for directing the COVID-19 genomic RNA translation into a structural and non-structural protein. Additionally, stem-loop structures within 5’-UTR and a leader sequence of the COVID-19 genome are essential for transcription and replication. Furthermore, the transcription regulatory sequence (TRS) located throughout the COVID-19 genome is one of the most distinguishing characteristics of COVID-19 and plays a crucial role in transcription regulation (Figure 2) [17].

### 2.2. Influenza A (H1N1)

The influenza virus has several antigens divided into three types of influenza A, B, C based on the antigenic disparity between nucleoprotein and protein matrix [18]. Based on the surface structure, hemagglutinin and neuraminidase, the B and C types of the influenza virus have merely one serotype [19], while the influenza A virus has 16 serotypes based on hemagglutinin and nine serotypes based on neuraminidase. In this vein, it should be stated that the highest incidence of the flu in humans is caused by the influenza A virus [20]. This virus can infect mammals and birds, including horses, pigs, and various species of poultry as well as humans [18].

The influenza A virus is a dangerous virus with a single-stranded, eight-piece RNA genome whose genetic structure lends itself to modification. Genetic mutations may be of minor nature, that may result in a local influenza epidemic in winters due to the widespread viruses, or a more extensive nature, that may give rise to the emergence of a novel virus creating more severe pathogenesis and causing a global epidemic [21]. Antigenic shifts occur only in influenza A and refer to the emergence of a new influenza virus that allows human-to-human transmission. In this context, international travel is a serious global health threat and should be restricted [22]. Influenza A genetic material is packaged into eight negative-sense single-stranded RNA in an enveloped virus. Neuraminidase (NA), hemagglutinin (HA), and less frequently, the matrix 2 (M2) protein are central membrane structural proteins in influenza A. Additionally, the matrix 1 (M1) protein is located beneath the membrane and is surrounded by eight vRNAs that exist as separate viral ribonucleoprotein (vRNP) complexes. In addition, M1 is a key mediator in viral morphology and is important for cell membrane binding. Among other structural proteins, HA is the main contributing factor in influenza A infection. In other words, HA with terminal sialic acid (SA) residues bind to sialylated receptors on the potential host cells [22].

## 3. Epidemiology

### 3.1. Epidemiological Prospective of COVID-19

China was the first country to report a case, and it had a higher morbidity and mortality rate than the rest of the world. Furthermore, up to 16 March 2020, Iran, with 14,991 patients, was the third country with the highest reported cases after China and Italy [23]. However, since 28 March, the United States has seen more than 590,000 COVID-19 patients and 24,000 deaths, far above the number of cases reported. COVID-19 cases are rising every day in Brazil, with the number of positive cases expected to exceed [24]. As of February 2021, more than one-hundred million confirmed cases worldwide of COVID-19 had been reported to the World Health Organization [25].

Following the increasing rate of the incidence and the global spread of the virus, the WHO issued a declaration on 30 January 2020, announcing the outbreak of the new coronavirus as the sixth cause of public health emergency of international concern, regarded as a threat not only to China but to all countries. Before this new coronavirus, the WHO had announced the last public health emergency for the outbreak of influenza H1N1 in 2009. The WHO officially named the new coronavirus COVID-19 on 11 February 2020 [26]; on the same day, the International Committee on Virus Taxonomy (ICTV) changed the name of the virus that causes this disease from 2019-nCoV to SARS-COV-2 [27].

#### 3.1.1. Death Rate by Age Group

The death rate and the probability of dying if infected by the virus are highly variable by age in COVID-19. The exploration of mortality rate by age in COVID-19 disclosed age-dependent mortality, and most deaths were recorded in patients older than 50 years. Furthermore, according to a new study conducted by UK researchers using data from mainland China, the average mortality rate of COVID-19 is 0.66%, rising sharply to 7.8% in people over 80 and falling to 0.0016% in children aged nine and younger. They also discovered that about one out of every five people over the age of 80 infected with COVID-19 would likely need hospitalization, compared to only 1% of people under 30 [28]. A study by Eduardo estimated the real-time case rate across different age groups by gender in Latin America (Table 1) [29].

#### 3.1.2. Death Rate by Gender Ratio

A sex bias with males > females was reported for COVID-19 by national case fatality rates (CFRs). Furthermore, SS Bhopal et al. examined the mortality ratio per 100,000 cases and concluded a higher fetal death rate among men than among women [30]. The most likely reasons for the variations found are hypotheses focused on risk factors that vary with sex and age. Differences in the profession, lifestyle (including smoking and alcohol consumption), physical comorbidities, and drugs are all examples. Rather than the science of genders, these theories are based on social and cultural considerations. Age, sex, and the previously listed risk factors, as well as gene expression and epigenetics, would all need to be taken into account when developing genetic bases [30,31]. Male sex is gradually being identified as a risk factor for COVID-19-related disease and death. Male dominance in COVID-19 mortality can be seen in almost all countries with sex-disaggregated records, in which males have a 1.7-fold higher chance of death than females [32]. Gender has a significant effect on immune cell transcriptomes; age affects immune cells or even the immune system differently based on sex. Aging causes a decrease in the proportion of naive T cells, which is more noticeable in males, while B cells decline mainly in males just after the age of 65. Between the ages of 62 and 64, male immune cells undergo abrupt and drastic changes in epigenetic landscape, resulting in an accelerated immunosenescence phenotype characterized by increased innate proinflammatory gene expression and lower gene expression related to adaptive immunity, which may potentially predispose older males to hyperinflammation and poor adaptive immunity [33].

#### 3.1.3. Death Rate by Health Conditions

Various underlying diseases have also been associated with an increased death rate. According to the Chinese Center for Disease Control and Prevention, respiratory infections, cardiovascular diseases, diabetes, cancers, and hypertension are major contributing factors for the increased death rate by COVID-19. Furthermore, an extensive cohort analysis of over 17 million people was held by EJ Williamson et al. to identify the clinical factors associated with COVID-19 related deaths. They identified hypertension as the most common comorbidity among COVID-19 related deaths. Their result can also be used for developing a predictive model [34].

### 3.2. Epidemiological Prospective of Influenza A (H1N1)

Influenza strikes at different intensities every year. Its epidemiological trend is influenced by a number of factors, namely the virus’s antigenic properties, transmissibility, and community susceptibility [35]. This virus can change the antigen properties of surface glycoproteins, hemagglutinin, and neuraminidase regularly. Major changes in these proteins are called ‘antigenic change’, and minor changes are called ‘antigenic shift’. Epidemics of influenza A are followed by antigenic variations, while local outbreaks are caused by antigenic modifications [36].

The influenza A virus caused several pandemics in the 20th century, so that millions of people have died from this virus in the last century [37]. The influenza A/H1N1 virus was first reported in April 2009 in Mexico and some states of the US. It was then rapidly spread in most countries, so that the WHO announced its pandemic in June 2009 [38,39,40]. Studies in Portugal [41] and Saudi Arabia [42] showed that the influenza A/H1N1 virus spread was 54.40% and 28.4%, respectively. This virus has been created from the simultaneous infection of pigs by the prevailing influenza A subtypes and the simultaneous amplification and shift of its genome, which has caused it to be more pathogenic than the other seasonal subtypes [43,44]

#### 3.2.1. Death Rate by Age Group

COVID-19 and influenza A pandemic differ in the age profile of critically ill patients and patients who die. The age distribution of morbidity during the 1918, 1957, 1968, and 2009 H1N1 pandemics was close to that of seasonal epidemics, with the majority of patients being under 60 years old. However, the percentage of under-60s among influenza deaths was significantly higher during the 2009 pandemic (peak 20 years) than seasonal epidemics [45]. In addition, based on an analysis of 153 influenza-associated deaths younger than five years old, N Bhat et al. suggest that the average influenza-related mortality rate for children was 0.21 deaths per 100,000, with the highest rate for children under the age of six months with a general reduction as they grew older [46].

#### 3.2.2. Death Rate by Sex Ratio

In most nations, males and females have equal case fatality rates for avian influenza. Human A (H7N9) cases, on the other hand, have a high degree of sex inequality, with males over 50 years of age having a slightly higher mortality rate [47]. In this background, T Paskoff et al. studied sex-based variations in mortality during the 1918 influenza pandemic on the island of Newfoundland in 2018 and discovered that mortality rates were generally equivalent in both sexes [48]. Contrary, adult males (15–44 years of age) in the United States and 12 other countries died at greater rates during the 1918 H1N1 pandemic [49].

#### 3.2.3. Death Rate by Health Conditions

C Rodríguez-Rieiro et al. in Spain identified a close association between comorbidity and influenza mortality rate. They also introduced asthma as the most prevalent risk factor among those with pH1N1-related ICU admission [50]. Additionally, N Bhat revealed that underlying diseases, such as neuromuscular disorders and chronic pulmonary disease, could increase childhood mortality from influenza [46].

## 4. Transmission and Replication

### 4.1. Transmission and Replication of COVID-19

#### 4.1.1. Transmission

Transmission efficiency was approved as a critical factor in the epidemiology of newly evolved viruses such as influenza A and COVID-19. Transmission modes, including nasal and oral droplets, indirect contacts, and a lesser degree, touching a surface with the virus on it, are accepted in COVID-19. Pharyngeal shedding is very high during the first week of symptoms and could enhance the risk of transmission. In addition, the presence of COVID-19 in stools and wastewater suggests its fecal-oral transmission. During the incubation cycle, Qun Qian et al. report clear signs of successful COVID-19 replication in a patient’s rectum, which may clarify COVID-19’s fecal–oral transmission [51].

Asymptomatic and pre-symptomatic transmission, a typical characteristic of infectious diseases, is possible in COVID-19 and contributes to its outbreak. Bai et al. reports a confirmed laboratory COVID-19 case without any clinical sign and suggests that asymptomatic cases could transmit COVID-19 [52]. In addition, Chen Yi et al. indicated that the infection rates of symptomatic and asymptomatic infections among close contacts were 6.30% and 4.11%, respectively [53]. This novel finding suggests the asymptomatic transmission of COVID-19. Growing evidence reveals that due to interaction mechanisms, the transmission and epidemiology of COVID-19 are dependently associated with influenza (Figure 3). ACE-2 overexpression resulting from a viral respiratory infection, in particular, influenza, was identified recently [54]. The study of M Domenech de Cellès et al. with a focus on the impact of influenza on COVID-19 transmission indicated that the transmission of COVID-19 has increased 2–2.5-fold in influenza co-circulating. This novel finding highlights the inevitable effects of influenza vaccination on COVID-19 mortality and transmission [55].

A number of days between virus infection and the onset of clinical manifestation are known as the incubation period. The incubation period is a key element in determining the quarantine period and could necessarily reduce epidemic size. The incubation period is around 14 days for COVID-19, meaning that a 14-day quarantine period is required for the complete absence of disease among healthy individuals exposed to the virus [56].

#### 4.1.2. Replication

Specific coronavirus spike (S) protein binding to the appropriate cellular entry receptors initiates coronavirus infection. Upon cell entrance, ORF1a and ORF1b are immediately translated into non-structural proteins. Here, prior to the replication process explanation, we describe the biological function of each NSP in the context of COVID-19 pathogenesis.

The C-terminal domain of COVID-19 nsp1 triggers translational inhibition by sterically blocking the ribosomal mRNA channel entry site in 43S pre-initiation complex, free 40S subunits non-translating 80S ribosomes. 5′ UTR of COVID-19 with a complex secondary structure efficiently promotes translation under a low ribosome. Therefore, upon COVID-19 infection, host protein production switches to viral protein synthesis [57]. Compared to other NSPs, nsp2 shows maximum sequence variability among coronaviruses, so it was believed that nsp2 protein coevolved in parallel with the hosts to obtain host-specific functions. Nsp2 and nsp3 are involved in RTC formation. Nsp3, through its post-translational activity, affects the host protein’s overall structure and; therefore, suppresses the host’s innate immune response [58].

A cytoplasmic double-membrane vesicle, required for COVID-19 replication, is induced by the cooperation of nsp4 and nsp3. Nsp5, also called 3C-like protease, or the main protease is thought to engage in synthesizing viral proteins and several nonstructural viral proteins through its protease activity. Moreover, nsp5 prevents interferon I signaling processes through cleavage of STAT 1 transcription factor [59]. By influencing autophagic proteins, including PIK3C3 and ATG5, nsp6 serves as an autophagy-inducing protein. Nsp8, in cooperation with nsp7, forms a heterodimer structure and confers RNA-binding capacity to nsp12. Nsp9 is required for binding coronavirus replication complex to RNA. Nsp10, an essential cofactor for 2′ O-ribose methyltransferase (Nsp16) and guanine-N7 methyltransferase (Nsp14) activation, assists in the methylation of mRNAs guanosine cap to promote transcription, splicing, polyadenylation, and nuclear export of viral mRNA. Nsp11 displays endo-ribonuclease activity that has a major impact on the viral life cycle [60]. Nsp12, an RNA-dependent RNA polymerase (RdRp), functions primarily in COVID-19 genome replication and transcription [61]. Nsp13, Helicase (Hel) or Nucleoside-triphosphatase (NTPase), facilitates RNA folding in the presence of NTPs and plays a crucial role in replication [62]. Nsp14 and nsp16, which stand for N-7- and 2-Oʹ-methyltransferase, respectively, are involved in the viral RNAs capping so that they are required for immune protection against host pattern recognition receptors (PRR), such as IFIT1. Notably, COVID-19 nsp13 (helicase), nsp14 (exonuclease), and nsp15 (endoribonuclease) were found to be highly effective viral interferon antagonists [63]. In conclusion, it is worth mentioning that NSPs are promising therapeutic targets for antiviral drug development.

Viral replication organelles (RO) are generally distinct functional structures indicated to function as ideal platforms for viral RNA synthesis. Double-membrane vesicles (DMVs) are the RO’s most abundant component and the central hubs for COVID-19 genome synthesis. By protecting the viral genome from innate immune sensors stimulated by dsRNA, DMVs create an appropriate condition for viral RNA synthesis. Newly synthesized viral RNA translocation from the DMVs to the cytosol is mediated by multiprotein complexes that span both DMV membranes. Therefore, these channels are critical for the COVID-19 life cycle. DMV molecular pores are principally made up of nsp3, nsp4, and nsp6 [64].

RTC contains nsp12-nsp7-nsp8 function in COVID-19 genome replication and transcription. COVID-19 replication is a complex process that involves RNA synthesis, proofreading, and capping. The COVID-19 genomic replication begins with the full-length negative-sense genomic synthesis that serves as a template to make a new positive-sense genomic RNA. In this context, the negative-strand synthesis is initiated by RdRp protein binding to the 3’ end of the genome in a process stimulated by 3’ end RNA secondary structures and sequences. Then, RdRp selects and binds to an appropriate nucleoside triphosphate (NTP) via the formation of phosphodiester bonds, so nucleotide incorporation results in the 3’ end of the nascent RNA extension. Processivity is critical for successful COVID-19 genome replication. Nsp7/nsp8 protein through interaction with the RNA backbone bestows high processivity on RdRp. The increased mutation rate in RNA virus replication is correlated with enhancing diversity in viral genomic sequence. Nsp14 provides a 3′–5′ exonuclease activity that assists RNA synthesis with a unique RNA proofreading function. Following replication, COVID-19 genomes need to undergo capping at their 5’ end and polyadenylation at their 3’ end. This process is similar to those that take place in eukaryotic cells, but COVID-19 genome capping and polyadenylation occur in the cytoplasm. Several momentous biological activities, including host immune response evasion, mRNA translation, and stability, have been reported for the final viral RNA cap [65].

### 4.2. Transmission and Replication of Influenza A

#### 4.2.1. Transmission

Several possible routes, including nasal and oral droplets and indirect contact, have been approved for Influenza A transmission. Indirect contact refers to transmission via a fomite, an object like a doorknob or toy, contaminated with the infectious virus. Additionally, airborne virus transport on microscopic particles, or the so-called aerosolized fomites, is also considered for Influenza A transmission [66]. Unlike COVID-19, effective transmission of influenza has not been observed in truly asymptomatic people. The incubation period is around 1.4 days for Influenza A, meaning that a 1.4 day quarantine period is required for the complete absence of disease among healthy exposed individuals [67]. Additionally, reproduction number (R0), an epidemiological transmission index in infectious diseases, is around 1.5 and 3 for influenza A and COVID-19, respectively [68].

This finding indicates that COVID-19 is more transmissible than influenza A. Indeed, different policies are required to control these diseases.

#### 4.2.2. Replication

Each vRNP comprises viral RNA (vRNA), viral polymerase, and a number of nucleoproteins (NP) molecules in influenza A. Two distinct conserved regions in 5′ and 3′ UTRs mediate the helical hairpin formation and allow RNA-dependent RNA polymerase (RdRp) to bind to vRNA. Contrary to most RNA-containing viruses, influenza viruses are transcribed and replicate in the nucleus. In this context, following the cytoplasmic appearance of the vRNA, the nucleocytoplasmic transport system, including importin-α and importin-β, detects nuclear localization signal (NLS) and transport vRNA into the nucleus. RdRp, as a heterotrimeric enzyme with three subunits, including PB1, PB2, and PA, plays a central role in replication and transcription. It is worth noting that many viruses exploit the perturbation of cell cycle progression to create a favorable condition for virus replication. RhoA kinase can potentially promote G1/S phase transition through pRb phosphorylation. By binding to and inhibiting NS1, H1N1 non-structural protein 1 (NS1) induces cell cycle arrest and favors cell condition for virus replication [69].

Two-step replication of influenza is initiated by complementary RNA synthesis (cRNA). In other words, prior to vRNA replication, vRNA is transcribed to cRNA. Following cRNA synthesis, newly synthesized nucleoprotein (NP) molecules and a single copy of the viral polymerase bind to cRNA to assemble into a cRNP. In a similar manner to cRNA synthesis, cRNA in cRNP acts as a template for vRNA production. In a sequence identical to cRNP formation, newly synthesized vRNA binds to accessory proteins, forming vRNP. Then, the influenza virus NS2 mediates vRNP nuclear export [70]. H1C as a member of the histone H1 family mediates intra-chromosome compaction. Besides, H1C promotes IFN-β production through IRF3 nuclear transportation. It was ascertained that H1C affects influenza virus replication by interaction with NS2. An extensive study by M Jin et al. demonstrated better replication of the H1N1 influenza virus in H1C knockout A549 cells than the wild-type A549 cells. As such, H1C plays a key role in H1N1 influenza host adaptation [71,72]

Notably, recent exploration by Liping Song et al. disclosed the relationship between miRNA and H1N1 influenza replication. Using a 3’-UTR reporter assay and virus proliferation analysis in MDCK cells, they identified putative miRNA complementary sequence in the influenza virus genome. They found that miR-323, miR-491, and miR-654 play a crucial role against influenza A infection via binding to the PB1 gene and, subsequently, inhibiting influenza A replication [73].

## 5. Clinical Manifestation

### 5.1. Clinical Manifestation of COVID-19

Twelve surface receptors of COVID-19, including ACE2, were determined by Gu et al. Among them, ASGR1 and KREMEN1 govern COVID-19 infection independently, so they may be specific receptors COVID-19 infection. These multiple host cell surface receptors enable COVID-19 to infect numerous body organs. Hence, COVID-19 patients present highly variable clinical symptoms in the clinic. Notably, no specific clinical signs that could reliably distinguish between COVID-19 and other viral respiratory infections have been reported. Most COVID-19 patients are mildly ill and experience flu-like upper respiratory symptoms, such as cough, shortness of breath, fever, dryness, fatigue, and so on. However, some patients may progress to acute respiratory distress syndrome, metabolic acidosis, or coagulation dysfunction and should receive mechanical ventilation and support in an intensive care unit (ICU) [8,9,17,22]. In addition, COVID-19 is associated with gastrointestinal symptoms, such as diarrhea, vomiting, or abdominal pain during the early phases of the disease (see Table 2 for a list of main symptoms). In the following section, we clarify the pathogenesis of COVID-19 in several organs.

#### 5.1.1. Cardiovascular

ACE2 is a membrane-bound aminopeptidase that plays a crucial role in the cardiovascular and immune systems [88]. ACE2 mediates COVID-19 entrance into specific host cells by binding to COVID-19 spike protein [74]. It was recently disclosed that people who suffer from cardiovascular disease experience a more severe disease than normal individuals due to ACE2 upregulation [75]. Thus, cardiovascular comorbidity contributes to elevated COVID-19 mortality. Furthermore, there is accumulating evidence that COVID-19 itself could trigger cardiovascular disorders, including venous thromboembolism, myocardial injury, and acute coronary syndrome (ACS). Cardiac troponin I is a cardiac regulatory protein that signals myocardial necrosis. Qing Deng et al. recognized elevated expression of Cardiac troponin I in COVID-19 patients during hospitalization.

Additionally, cardiac troponin I level was significantly higher in severe cases than in non-severe cases. In this manner, COVID-19 could function as a potential contributing factor to the development of myocardial damage [89]. Together, these data support bidirectional causation between COVID-19 and the cardiovascular system.

#### 5.1.2. Gastrointestinal (GI)

ACE2 expression is greatly more abundant in the human and mouse small intestine than all other organs, such as lungs. In addition, coordinated action of TMPRSS2 and TMPRSS4 significantly increases COVID-19 infection in human small intestinal enterocytes. GI symptoms, including diarrhea, nausea, and vomiting, occur in COVID-19 patients, but the most frequent symptom is the lack of appetite. The incidence of gastrointestinal complaints was greater in 19 patients hospitalized in medical units and intensive care units (ICU) than in patients observed solely in the emergency care unit [90]. Besides, M. Aziz et al. investigated the role of GI symptoms in predicting the severity of COVID-19. They indicated that GI symptoms, especially diarrhea, is associated with worse outcome [91]. All these findings suggest that COVID-19 is actively involved in GI tract infection.

#### 5.1.3. Lung

Ling Leng et al. used a comprehensive proteome study and bioinformatics analysis to discover proteomic profiles of COVID-19-infected human lung tissues. In the COVID-19 lung tissue, ferroxidase ceruloplasmin (CP), which is associated with the peroxidation of Fe (II) transferrin to Fe (III) transferrin, was significantly elevated resulting in increased oxygen intake, which affected normal relaxed breathing. Increased SLC4A1 expression was also found in COVID-19 lung tissue, which resulted in a malfunction of gas exchange in the lungs as well as urinary acidification. SFTPB is associated with the development of lamellar bodies in the alveoli as well as the removal of surface tension. The COVID-19 lung tissue displayed a significant decrease in SFTPB. In addition, COVID-19 lung tissues have a high level of activation of the non-canonical NF-B/NFKB2 pathway, which is a central mediator in cytokine and chemokine synthesis. It is worth noting that the non-canonical NF-B/NFKB2 pathway has never been confirmed to be activated during cytokine storms caused by other respiratory viruses, including influenza [92].

### 5.2. Clinical Manifestation of Influenza A (H1N1)

Influenza A has been recognized as a major contributing factor in acute respiratory tract infection. Additionally, pharyngitis, dyspnea, diarrhea, nausea/vomiting, headache, shortness of breath, runny nose/rhinorrhea, Sore throat, and myalgia are also frequently seen in influenza A patients [77]. On the other hand, high detection of influenza A (H1N1) in myocardial and pericardial tissues and fluid implies a causal relationship between influenza A (H1N1) and myocarditis [93]. It has been confirmed that influenza A (H1N1) viruses infect and subsequently induce GI cell death via sialic acid (SA)–α 2, 6–galactose (Gal)-terminated saccharides [94]. A Riquelme et al. retrospectively conducted a study on the prevalence of GI clinical symptoms in Chilean influenza A (H1N1) patients. They observed 72 patients with vomiting (14.3%), 147 patients with diarrhea (29.4%), and 182 patients (36.4%) with nausea. Besides, gastrointestinal symptoms were observed in one out of four influenza A (H1N1) patients in a similar study in North America [78].

Notably, a significant accumulation of viral replicative intermediate agents such as dsRNAs after infection with influenza A (H1N1) were noticed, which were recognized by innate immune receptors such as membrane-bound TLR3 and cytosolic RIG1-like receptors (MDA5, RIG-1, and LGP2). MDA5 appears to be a Runx3 inducer, which is intriguing. Cell apoptosis is thought to be a cellular process that effectively clears virus-infected cells, thereby acting as a powerful tool for the prevention of virus spreading. An excessive amount of uncontrolled apoptosis can trigger pulmonary tissue damage and lung dysfunction, which would increase morbidity and mortality. Runx3 is known to modulate the H1N1 influenza-induced host airway epithelial cell apoptosis. Runx3 overexpression in airway somatic cells, induced by H1N1 influenza infection, aggravates the disease severity by promoting host cell apoptosis and tissue injury [95].

We finally investigate two important questions as to viral infection as follows:

#### 5.2.1. How Obesity Impacts Viral Infections?

COVID-19 has a greater effect on ACE2 expression in adipose tissue than in lung tissue, which is a significant target tissue. Obese people have more adipose tissue, which means they have more ACE2-expressing cells and, as a result, obesity can increase infection susceptibility and a risk factor for COVID-19-related mortality [80]. In a systematic review of the literature, J Siqueira et al. determined a close association between increasing BMI and deterioration in clinical outcome. They concluded that obesity led to higher levels of hospitalization, poor outcomes, and increased mortality in COVID-19 infected patients [96].

Additionally, a close correlation was recently found between obesity and the risk of intensive care unit (ICU) admission and hospitalized patients with influenza A (H1N1) infection. L. Fezeu indicated that severely obese influenza A (H1N1) patients had a twofold higher risk of mortality and ICU admission compared with influenza A (H1N1) patients who were not severely obese [79].

#### 5.2.2. Why Women Have Less Tendency to Be Affected by Viral Infections?

A growing body of evidence suggests that many variations exist between men and women in the immune response to infection. Data from hospitals worldwide disclosed that respiratory system diseases, such as those triggered by acute viral infections, are more prevalent in men (25%) than in women. The presence of two X chromosomes in women affects the immune system even if one is inactive. Several immune system components, including TLR7, FOXP3, CD40L, TLR8, and CXCR3, are affected by the X chromosome and can be upregulated in women and impact the response to viral infections. It has been proposed that the X chromosome encodes several immune regulatory genes, and they could trigger lower viral load levels, inflammation, and death after viral infection in females than in males. The TLR7 gene located on the X chromosome is expressed in numerous immune system cells, such as B cells, macrophages, dendritic cells, and circulating monocytes. TLR7 plays a critical role in single-strand RNA virus recognition via inducing anti-COVID-19 antibodies and generating pro-inflammatory cytokines, including IL-1 and IL-6 family members. As such, the TLR7 overexpression in women is associated with enhanced resistance to viral infections [97].

Furthermore, plasma levels of ACE2 were compared in 1485 men and 537 women with heart disease from two different cohorts. It was discovered that men had higher plasma ACE2 levels than women. The study indicates that the variation in plasma ACE2 can explain the intensity of COVID-19 in men [98].

## 6. Diagnosis

### 6.1. Diagnosis of COVID-19

In the currently evolving COVID-19 pandemic, it is crucial to rapidly and reliably identify the cases to begin therapy and minimize transmission. Due to the possibility of asymptomatic transmission in COVID-19, it is important to develop a rapid and highly sensitive protocol for the accurate determination of asymptomatic people. Following are some major tests that are used to diagnose this virus.

#### 6.1.1. Molecular Tests

By revealing the complete genome of COVID-19 and significant advances in RNA methodologies, the molecular assay can be applied for COVID-19 diagnosis. Among them, real-time RT-PCR is widely employed in diagnostic virology and is considered a gold standard for the detection of some viruses. Previous studies disclosed that there are many clinically diagnosed COVID-19 patients in accordance with clinical manifestation and CT scan, but the PCR test shows a negative result. In this manner, a negative PCR test result does not rule out COVID-19 infection, and the PCR assay along with CT scan and clinical symptoms should be considered in clinical decision-making during the COVID-19 pandemic. Several factors, including improper transportation, collection or handling, and amplification inhibitors in the sample, can contribute to false-negative results.

Additionally, real-time RT-PCR results are also affected by the period of disease development and sampling timing. High-quality extraction and the real-time RT-PCR kit could greatly enhance the approach and reduce inaccurate results [99]. False-negative and false-positive results of real-time RT-PCR tests should be considered important in final decisions. As soon as first sequence information becomes available, a real-time RT-PCR specific for influenza A (H1N1) virus is set up and evaluated thoroughly from a technical and clinical aspect [83].

Recently, newly innovative diagnostic methods based on Clustered Regularly Interspaced Short Palindromic Repeats (CRISPR) gene-editing technology were developed to detect nucleic acids. The SHERLOCK COVID-19 detection protocol, which is composed of three steps, only lasts 1 h. Despite qPCR, the SHERLOCK COVID-19 detection protocol is an extremely powerful technique for specific DNA/RNA detection at low attomolar concentrations [100]. In this context, Cas13a is RNA-specific, but Cas12a works with DNA. Here, in order to improve the accuracy of the assay, targeted DNA or RNA amplification by recombinase polymerase amplification (RPA) or reversed transcriptase-RPA (RT RPA) occurs prior to initiating the reaction [82].

#### 6.1.2. Computed Tomography (CT) Scan

Early detection of COVID-19 by chest CT scan with high accuracy is more possible now. A retrospective analysis by Chunqin Long showed that the sensitivity of initial CT was 97.2% [81]. Imaging technology has become a powerful tool in severity assessment and predicting disease progression of COVID-19 infection [101]. A close similarity in chest CT findings was made between influenza A and COVID-19. Furthermore, careful examination illustrates dissimilarity in chest CT between influenza A and COVID-19, in which influenza A ground-glass opacity has a central, peripheral, or random distribution as well as five affected lobes.

In contrast, COVID-19 ground-glass opacities have frequently been placed in the periphery of lower lobes. Additional characteristics, including pleural thickening vascular engorgement and subpleural lines, were also observed in individuals suffering from COVID-19. On the contrary, pneumomediastinum and pneumothorax are merely seen in influenza A patient. Therefore, these few differences enable physicians to distinguish between influenza A and COVID-19 [102].

#### 6.1.3. Lymphopenia

Lymphocytes are essential for immune homeostasis and the inflammatory response in the body. Lymphopenia is a condition in which blood lymphocyte count is decreased and reaches below 1100 cells/μL (≤1100 cells/μL). This condition is frequently seen in infectious diseases, such as COVID-19. It was recently disclosed that COVID-19 results in a significant decline of lymphopenia, especially in younger patients. Thus, lymphopenia is an indicator of poor prognosis in COVID-19 [84].

On the other hand, influenza causes a reduction in lymphocyte count (<0.8 × 109/L) in adult patients. Additionally, Yandong Cheng in China identified lymphopenia in the severe form of influenza A infection. Thus, lymphopenia and other clinical parameters can be used in the early diagnosis of the severe forms of influenza A and COVID-19 infection [85].

#### 6.1.4. Gut Microbiome

The gut microbiome plays a crucial role in metabolism and host nutrition and greatly influences disease and human health. There is a close correlation between reduced antiviral immunity and intestine flora dysregulation [86]. An experimental study by Deriu et al. indicates that altered intestinal microbial profile occurs in influenza infectious [87]. These results imply the development of disease through the gut-lung axis. An exploration of gut microbiota in patients with influenza A, COVID-19, and healthy subjects was conducted by Silan Gu et al. They detected a decreased bacterial diversity and increased opportunistic pathogens in COVID-19 patients compared with healthy subjects. In addition, influenza A patients show reduced bacterial diversity and different gut microbiota than COVID-19 patients. Taken together, gut microbiota composition could function as a diagnostic biomarker in delineating between COVID-19 patients and influenza A patients [86].

### 6.2. Diagnosis of Influenza A (H1N1)

#### 6.2.1. Conventional Culture

This method takes around one to 14 days and allows the isolation of many viruses. In addition, unexpected or novel viruses can be detected, although some viruses do not grow in routine cultures. It is more sensitive than antigen detection [103].

#### 6.2.2. Antigen Detection

Since antibodies have a database of prior exposure and are a low-cost technology, antibody assays have proved effective in previous surveillance programs. The direct immunofluorescence assay (DFA) detects influenza virus epitopes using species-specific monoclonal antibodies. DFA takes only 1 to 2 h to complete, but it requires significant experience to provide correct results [103].

#### 6.2.3. Molecular Tests

In the diagnosis and transcription of influenza viruses, molecular diagnostic methods are useful. In comparison to traditional virus culture methods, molecular methods provide results in a short period of time. V Sharma et al. recently used standard influenza strains to standardize molecular diagnostic assays. On RNA isolated from normal strains, conventional one-step RT-PCR, Taqman real-time RT-PCR, and RT-LAMP (reverse transcription loop-mediated isothermal amplification) were standardized. They discovered that RT-LAMP is a fast, sensitive, specific, and cost-effective method for detecting and subtyping influenza viruses [104]. Furthermore, in 2010, P López Roa linked real-time RT-PCR to traditional cell culture for the detection of pandemic influenza A (H1N1) in hospitalized patients. They concluded that real-time RT-PCR had high sensitivity and accuracy for detecting influenza A H1N1 in adult patients [105].

## 7. Treatment

### 7.1. Availability of Treatment against COVID-19

A confirmed COVID-19 patient requires full bed rest and compassionate care, as well as sufficient calorie and water consumption to avoid dehydration. Antiviral drugs are an alternative strategy to prevent COVID-19 transmission and expansion. Amphotericin B (AmB) with a broad spectrum of antimicrobial properties categorized into polyene group. AmB via numerous processes, including increased macrophage phagocytic, TLR, IL-6, and TNF-α, enhance immunity. In the case of antiviral activity, it was recently proven that AmB could greatly influence cholesterol structure in viral envelopes and cellular membranes. These findings suggest AmB as a novel candidate therapeutic drug in COVID-19 [106].

M^pro^, which mediates COVID-19 replication, was recently identified as an attractive drug target in COVID-19. Virtual screening and high-throughput screening as an efficient tool for rapid drug discovery against COVID-19 were performed by Z Jin et al. They concluded that N3 and ebselen through targeting substrate-binding pocket of M^pro^ inhibited M^pro^ activity. These novel findings highlight N3 and ebselen as a therapeutic drug against COVID-19 [107]. Remdesivir, an approved drug against the Ebola virus, inhibits COVID-19 RNA synthesis by targeting the RNA-dependent RNA polymerase (RdRp), so it could act as a promising antiviral drug against COVID-19 (Figure 4) [108]. Favipiravir is a new drug used in COVID-19 treatment by converting into an active phosphoribosylated form and subsequently targeting RdRp. Chloroquine phosphate, interferon α (IFN-α), and ribavirin are other antiviral drugs useful in COVID-19-induced pneumonia [109].

Vitamin D is a micronutrient that is urgently required for maintaining skeletal health, and its deficiency could greatly influence bone performance. Vitamin D emerged as a key player necessary for a well-functioning immune system and could enhance viral infections’ immune response. Several studies have revealed the close association between decreased levels of vitamin D and viral disease, including influenza A and COVID-19. Association between COVID-19 related death and vitamin D deficiency were reported recently. E. Laird et al. indicated that vitamin D deficiency increased COVID-19 mortality rate. Finally, a growing number of studies throughout the world propose vitamin D as an efficient prophylactic agent for COVID-19 disease [109].

Convalescent blood products (CBP) as an effective approach against pathogens can be divided into numerous subgroups, including whole convalescent blood (CWB), high-titer human Ig, convalescent plasma (CP), and polyclonal or monoclonal antibodies. Passive immunization, which refers to the transfer of active humoral immunity via CP, plays a central role in preventing and managing infectious diseases. CP composition includes water, inorganic salt, and a growing number of different proteins such as complement factors, immunoglobulins, and antithrombotic factor. Neutralizing antibody (Nab) is a key functional component of CP that binds to S proteins of COVID-19 and is involved in CP antiviral activity. It was recently recognized that anti-inflammatory cytokines and antibodies are responsible for CP’s immunomodulatory property due to their role in blocking inflammatory cytokines and autoantibodies [110].

A prospective study with 10 severe COVID-19 patients was performed by Kai Duan et al. in China, in which transfusion of high neutralizing antibody titers CP was applied in seriously ill patients to improve antiviral activity. Within three days of CP transfusion, improvements were observed in clinical symptoms and laboratory parameters along with the safety of the CP transfusion. They also identified elevated levels of neutralizing antibody and lymphocyte count and markedly decreased C-reactive protein. In extreme COVID-19 cases, this study found that CP therapy was well tolerated and could increase clinical outcomes by neutralizing viremia [111].

Cytokine is associated with excessive cytokine production in response to the presence of several stimuli, such as a viral infection. Acute respiratory distress syndrome (ARDS) is considered a major contributor to the pH1N1 and COVID-19 related death. Cytokine is regarded as a primary cause of ARDS and multiple-organ failure. Multiple clinical examinations have detected a cytokine in critically ill patients with COVID-19. Thereby, effective suppression of the cytokine could functionally prevent the deterioration of COVID-19 patients and save the patients’ lives [112].

MSC has numerous unique features, including antimicrobial activity, immunomodulatory, and regenerative properties, as well as an immune-privileged phenotype, making it an attractive therapeutic option for COVID-19. MSCs are a well-established producer of anti-inflammatory chemokines and cytokines, including transforming growth factor-beta, IL-10, and prostaglandin E2 (PGE2). Additionally, MSCs have been shown to effectively antagonize the release of certain pro-inflammatory cytokines [113]. Several pre-clinical investigations have indicated that MSCs effectively work against ARDS and acute lung injury (ALI), both of which are severe clinical manifestations of COVID-19. At the same time, the administration of MSCs to ARDS patients was well tolerated [114]. By RNA seq analysis, Zikuan Leng et al. disclosed that ACE2 and TMPRSS2 are not expressed in MSCs, so MSCs infection with COVID-19 is not possible. Furthermore, following the administration of intravenous injection of MSCs, they reported a great improvement in the resolution of inflammation in severe COVID-19 patients [76].

### 7.2. Availability of Treatment against Influenza A (H1N1)

Antiviral drug strategy is now considered an efficient tool to restrict influenza infection. Recent work suggests that antiviral therapy needs to initiate within 48 h of the first clinical symptom for effective management [115]. Oseltamivir and zanamivir are newly introduced drugs that act through NA inhibition. Furthermore, amantadine and rimantadine have exhibited promising performance against M2 ion channel activity commonly used in influenza disease [116]. Interferon-inducible trans-membrane (IFITM3) protein family, which belongs to interferon-stimulated genes (ISGs), restricts influenza A cell entrance. Thus, IFITM3-induced drugs with an influenza-restricting activity have become a new candidate for influenza A [117].

Additionally, in a study by H Khalili et al. in Iran, vitamin D concentration was evaluated in patients infected with H1N1. They revealed that 60.8% of patients were deficient in vitamin D [118]. Consequently, vitamin D can prevent the spread of both pH1N1 and COVID-19 by boosting our immune system. Furthermore, in pH1N1, a reduction in mortality, serum cytokine response, and respiratory tract viral load were detected in patients who underwent CP transfusion [119]. Consequently, these novel findings introduce CP therapy as a candidate therapeutic option for H1N1 influenza.

## 8. Prevention and Control

### 8.1. Prevention and Control of COVID-19

So far, prevention has been the best way to reduce the effect of COVID-19 and people’s infection with it [120]. People should be informed about and use the latest COVID-19 outbreak information provided by the World Health Organization and the Ministry of Health in each country. They should also avoid secondary infections and other diseases [121].

Some practical measures to control and prevent coronavirus infection are (1) using a face mask; (2) washing hands with soap regularly for at least 20 s and/or applying hand disinfectants containing at least 60% alcohol; (3) restricting trips and avoiding contact with infected people (people who cough or sneeze and are suspected of the infection); (4) keeping a safe distance from people (minimum distance of 1.5 m); (5) not touching eyes, nose, and mouth with unwashed hands [121]; (6) avoiding public places and large communities; and (7) visiting a physician if you have a fever or cough or severe breathing problems. In addition, according to the American Academy of Ophthalmology, contact lenses cannot be worn because they allow people to touch their eyes more often [122]. Cleaning and disinfecting regularly touched objects and surfaces is another COVID-19 avoidance technique [123]. Following is a layover to explain the prevention protocol against COVID-19 (Figure 5).

### 8.2. Prevention and Control of Influenza A(H1N1)

While influenza vaccine is generally well-tolerated, minor side effects such as soreness, redness, swelling, low-grade fever, and aches can occur but are uncommon. Fever is a rare side effect of the influenza vaccine, occurring in just 1% to 2% of those who receive it [124]. Currently, three types of vaccines (inactivated, live attenuated, and recombinant HA vaccines) are approved in different countries, each with its own set of benefits and disadvantages (high cost and poor productivity, side effects, and reduced use in people aged 18 to 49 due to low immunogenicity) [125]. Live attenuated virus vaccines generate good cellular and humoral immune responses, both systemically and at the mucosal level (immunoglobulin A) [126]. Comparisons of nonadjuvanted inactivated vaccines and live attenuated vaccines show that each can protect, with live attenuated vaccines having an advantage in infants and inactivated vaccines having an advantage in people who have had multiple previous exposures to influenza antigens.

Additionally, adding oil-in-water emulsion adjuvants to inactivated vaccines increases usable antibody titers, broadens antibody cross-reactivity, and reduces antigen dosage [127]. Recent work by RB Belshe [128] revealed that, in young children, the live attenuated vaccine had slightly greater effectiveness than inactivated vaccine. Adjuvants, which enhance the immune system while still having a low-risk profile, are one way to strengthen influenza vaccines. Alum, MF59, AS03, AF03, virosomes, and heat-labile enterotoxin are the six adjuvants currently used in licensed human vaccines (LT). In the event of a pandemic, the inclusion of an adjuvant will allow for quicker responses and dose sparing, resulting in better coverage. This work could be performed by any of the adjuvants mentioned in [129].

## 9. Vaccine Production Strategies

### 9.1. COVID-19

There is a global consensus that an effective vaccine is the most effective approach to controlling the COVID-19 pandemic sustainably. Extensive efforts and general coordination have contributed to the fast growth of vaccine production and the initiation of clinical trials throughout the world.

Nanotechnology systems have turned out to be excellent tools for the production of new and efficient vaccines, so that they have remarkably helped generate new vaccines fast, which will proceed to clinical trials [130,131]. Along with inactive vaccines, nanotechnology-based vaccines, such as mRNA-based vaccines that are carried by nanoparticles and the vaccines built by virus vectors, have already reached phases II and III of clinical trials [132].

### 9.2. Inactive and Live-Attenuated Vaccines

These vaccines contain inactive or live-attenuated viruses and are a classic strategy for viral vaccination. Examples include the human rabies vaccine, animal rabies vaccine, and oral polio vaccine [133].

### 9.3. Live-Attenuated Vaccines (LAVs)

These vaccines contain non-pathogenic amplifiable live viruses. LAVs are designed to produce immunity with only one dose without inducing the disease. The production of live-attenuated viral vaccines is challenging and needs extensive biological immunity. Since LAVs production technology is available and established, these vaccines are regarded as a leading candidate for COVID-19 vaccine production. However, LAVs have some drawbacks, such as virus transfer, conversion to pathogenic form, re-activity in people with weak immunity system, and recombination with related viruses circulating among people, especially new infections whose pathogenicity is still unknown. Furthermore, LAVs generally need a cold transfer chain. The loss of the virus effectiveness and its reproduction potentials during vaccine production are also important challenges [132,134].

Presently, novel technologies like genetic code expansion are used to produce highly amplifiable and genetically sustainable viruses. It is also possible to use genomic methods to produce and synthesize recombined SARS-COVID-2 from viral DNA fragments. These strategies can be employed for the fast production of LAVs [135].

Johnson and Johnson Company is experienced in producing the Ebola vaccine, known as Janssen’s AdVac^®^ adenoviral vector, by the PER.C6^®^ cell line technology. Since AdVac^®^ was satisfactory, the company is producing a COVID-19 vaccine based on its previous advanced technology. In Hong Kong University, a live influenza vaccine has been produced that expresses COVID-19 proteins [136]; in addition, Codagenix has employed the opcodon deoptimization II technology to develop a new strategy [137].

### 9.4. Inactivated Vaccines (IVs)

Viruses are inactivated by heat, chemicals, or a combination of these two. These vaccines cannot amplify, so they are safer than LAVs. It should; however, be noted that the inactivation operation reduces vaccine immunogenicity, so replicated doses are required to enhance the vaccine’s long-term immunogenicity. Although IVs are more stable than LAVs, they need a cold chain [138]. Several IVs are producing against COVID-19. For instance, the first clinical trials of the vaccine produced by Sinavac Company have been confirmed [132].

### 9.5. Designing Nucleic Acid-Based Vaccines

Genetic codes for the direct production of viral proteins are a promising alternative method for traditionally designing and producing vaccines. Both DNA and mRNA nucleic acid structures have been used in producing COVID-19 vaccine [132]. This vaccine production infrastructure has drawn attention to the vaccines’ production speed, stability, and immunogenicity. However, there are, unfortunately, numerous reports on the failure of these vaccines in clinical trials. So far, no RNA and DNA vaccines have gotten the license for human injection. Nonetheless, an applied advantage of these vaccines is high antibody production rate, CD4+T cell-specific immunity responses, and the stimulation of CD8+ cytotoxic T-cell responses, which play a key role in virus eradication [139,140,141].

To produce COVID-19 vaccine DNA, Inovio Pharmaceuticals, a pioneer company, performed phase 1 clinical trials on 6 April 2020. Entos Pharmaceuticals in Alberta, Canada, has also gone through phase 1 clinical trials [132].

mRNA-based vaccines are produced by in vitro transcription mechanism for which cell requirements and the barriers of transcription in the cell should be tackled. The mRNA production technology of Moderna Company reached phase 1 clinical trials on 16 March 2020, which had promising results. Also, bioNTech-Pfizer in Germany has passed phases 1 and 2 clinical trials of four mRNA-based vaccines [132,142]

Two vaccines that have received licenses in the US and UK are produced by Moderna and bioNTech-Pfizer. These two vaccines exhibited 94–95% effectiveness. However, new information will be available by further research over time. The Pfizer vaccine produces 52% immunity in the first dose, which reaches 95% after the second dose. It should be noted that the Pfizer vaccines should be stored at −70 °C, which requires specific freezers. Additionally, its storage time is only five days. However, the Moderna vaccine can be kept in a normal freezer (−20 °C) for one month. As a result, despite the similar effectiveness, the Moderna vaccine is requested more [143,144]

### 9.6. Viral Vectors

Some mammal viruses have been engineered to be used in vaccine production. Such a vaccine candidate is developing for COVID-19. CanSino Biological and Oxford University have used adenoviral vectors for the COVID-19 vaccine, including adenovirus type 5 (Ad-nCov) and chimpanzee adenovirus-based vector (ChAdOx1) since 16 March 2020 and 31 March 2020, respectively [132]. The adenoviral vectors have various advantages, including extensive tissue tendency, intrinsic adjuvant property, and scalability. A challenge in using adenoviral vectors is the high immunity levels, for example, against Ad5, as reported in numerous works [145,146]

Recently, the Oxford AstraZeneca vaccine was approved for injection. This vaccine’s effectiveness varies from 62% to 90%. This vaccine should be stored at 2–8 °C, and it can be kept for up to six months. A key advantage of this vaccine is its low price among the approved vaccines [147].

### 9.7. Protein Vaccines

The main protein part of the coronavirus is its surface spike region, which is very similar to the protein regions of SARS and MERS viruses. If this protein is combined with an immunogenic protein, the immunity response will be well stimulated. Now the question is that how this protein can be identified in practice, not in theory?

By achieving the serum of a corona-contracted person and identifying the antiviral antibodies by the display phase identification technique, the specific antibody sequences bonded to the virus are determined. After identifying the peptide sequence of the virus, which is attached to the antibody, amplifying this peptide fragment, and isolating it, it can be injected into healthy people to mimic the performance of the coronavirus and make the healthy people’s body synthesize the coronavirus antibody [148,149]. Following is the detail of vaccines that currently developed by different companies all over the world (Table 3).

## 10. Vaccine Production Strategies of Influenza A (H1N1)

Vaccination is considered to be an effective safety measure towards influenza contamination and ought to be presented throughout all healthcare visits and hospitalizations at any time throughout the influenza season. Vaccine efficacy is set at 60% in an excellent season; however, if the vaccine no longer fits the present-day circulating traces of the virus, effectiveness may be reduced to 10% to 20%. The basic anticipated effectiveness of influenza vaccines is 38%. It is usually recommended that anyone who is in six months or older than six months may get hold of an annual influenza vaccination until they have contraindications for the vaccine or any of its components. Ideally, the vaccine should be administered in the start of October as it takes approximately two weeks from vaccination to immunization. However, immunity from vaccination wanes over time, so selecting an appropriate time to get hold of the vaccine may be difficult due to the fact one cannot expect while influenza outbreaks will height throughout the season. The peak time for outbreak ought to arise early within the season (e.g., October) or later within the season (e.g., April or May). Therefore, if a man or woman gets a vaccine early within the season, they will have much less safety if the virus peaks later. The vaccination may be given simultaneously with different inactivated vaccines; however, it ought to be administered in separate anatomic sites [150,151].

The influenza A (H1N1) 2009, inactivated, Hualan Natural Bacterin Company created monovalent immunization, and the seed infection was produced from reassortant immunization infection A/Califor- nia/7/2009 NYMC X-179A (Modern York Restorative Col- lege, Unused York), disseminated by the Centers for Illness Control and Avoidance within the Joined together States. This strain was suggested by the World Health Organization and derived from the Chinese Nourishment and Sedate Administration [152].

### 10.1. Vaccines Produced against Influenza A (H1N1)

#### 10.1.1. Egg-Based Vaccines

Most producers use eggs to develop the flu infection during the advancement cycle. Hypersensitive responses should not be noticed in individuals with egg allergies for 30 min after the organization of the immunization. Notwithstanding, the CDC gives different direction to medical services clinicians, especially if the individual has a background marked by an extreme egg allergy [151].

#### 10.1.2. Attenuated Vaccines

The live, attenuated influenza vaccine, FluMist (AstraZeneca), which is controlled by means of a nasal splash, is likewise quadrivalent. This antibody is egg-based and might be regulated to anybody aged two to 49 years without contraindications. It has been observed that the risk of Reye’s syndrome in children receiving live attenuated vaccines and children taking medications with salicylate (such as aspirin) increases. Subsequently, individuals aged two to 17 years with contraindications should stay away from the live lessened flu vaccine [151].

#### 10.1.3. Recombinant Vaccines

Recombinant flu antibodies (RIVs) are made utilizing recombinant innovation. It is a method that modifies the hereditary fabric to enhance alluring characteristics. Rather than utilizing eggs, the cells are developed in cultured cells of mammalian origin. The benefits of RIVs over egg-based antibodies are that they can be created quicker, and the threat of pandemic, egg deficiency, or transformations can be suppressed [151,153].

## 11. Conclusions

Since the COVID-19 appearance in Wuhan in 2019, extensive attempts have been made to demonstrate this novel coronavirus’s features. This review study was conducted to increase our understanding of both COVID-19 and influenza A (H1N1). Many questions are open and need an answer, of which the most frequently asked one is how much COVID-19 differs from influenza A (H1N1) and whether COVID-19 has originated from influenza A (H1N1) or not. To answer these questions, different aspects of both influenza A (H1N1) and COVID-19 pandemic, including molecular biology, clinical symptom, diagnosis, epidemiology, and treatment, have been discussed. We compared clinical presentations between COVID-19 and influenza viruses and found similar clinical symptoms. That is, they both cause respiratory disease, cardiovascular disorder, GI disease, and acute lung injury. Since these clinical findings are common among viral and bacterial infections, they are not an appropriate criterion for the decision on the COVID-19 origin. On the other hand, we found that both influenza A (H1N1) and COVID-19 utilize a distinct cell entrance mechanism and genome replication.

Additionally, an extensive study aimed to determine the unique host gene expression signature response to COVID-19 infection compared to influenza A (H1N1), which gave us a more in-depth understanding of similarity and differences in host response performed recently. It detected several genes’ unique expression, including serum amyloid A 2 (SAA2), CSF2/3, and PTGS2 in COVID-19 infection [109]. This novel finding undermines our hypothesis that COVID-19 originated from influenza A (H1N1). However, further investigation is required to decide COVID-19 origination and evolution. In the future, we can have a large collection of data in a database, which can assist in studying the origin and evolution of coronavirus.

## Figures and Tables

**Figure 1 viruses-13-01145-f001:**
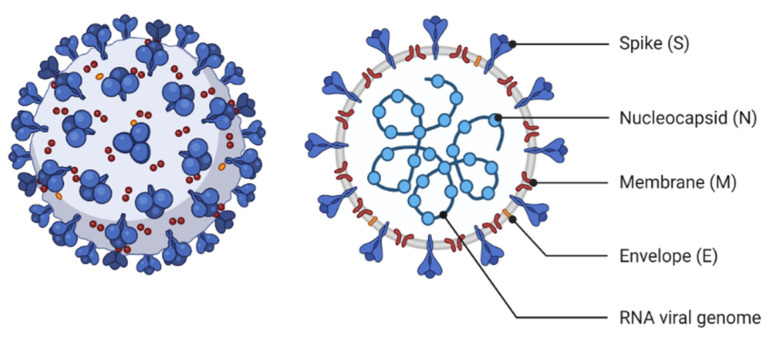
The structural characteristics of Coronavirus.

**Figure 2 viruses-13-01145-f002:**
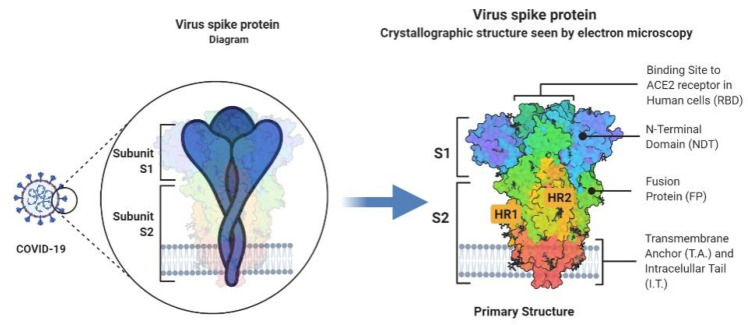
An in-depth look into the structure of the Covid-19 spike glycoprotein.

**Figure 3 viruses-13-01145-f003:**
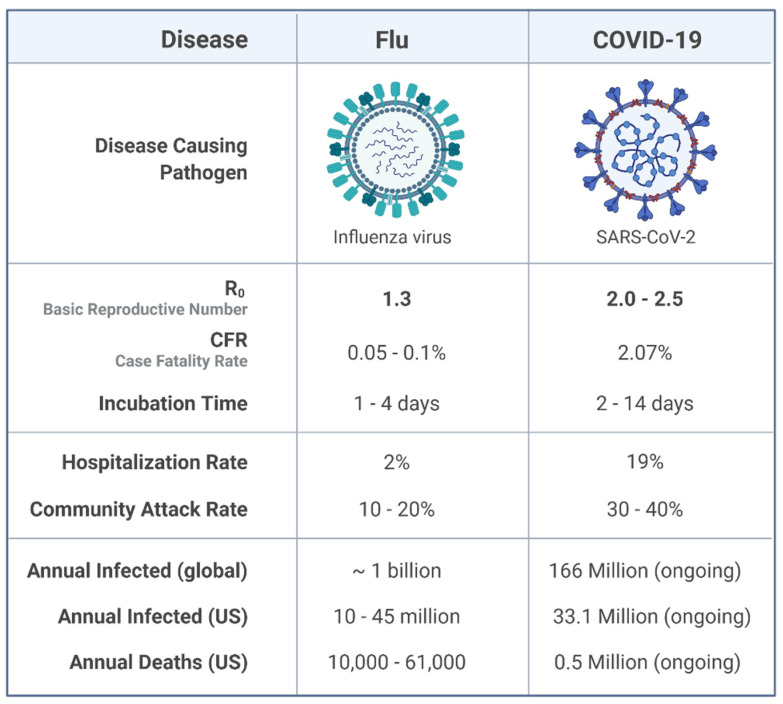
Epidemiological comparison of Influenza A and COVID-19.

**Figure 4 viruses-13-01145-f004:**
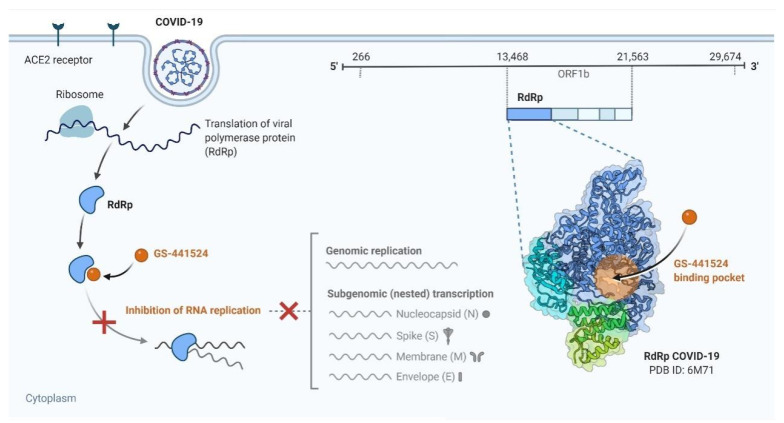
Remdesivir is a potential repurposed drug candidate for COVID-19.

**Figure 5 viruses-13-01145-f005:**
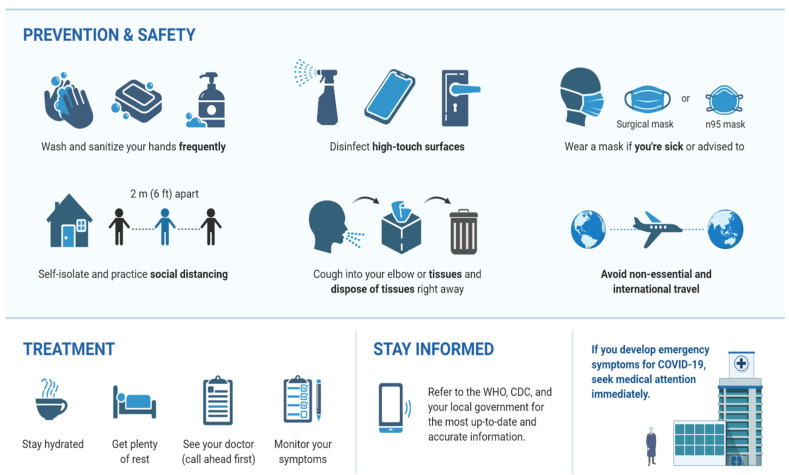
Prevention and safety protocol to avoid COVID-19.

**Table 1 viruses-13-01145-t001:** Ratio of COVID-19′s cases, deaths, and morbidity according to different age groups.

Gender proportion of COVID-19 cases	Age (Years)	10	20	30	40	50	60	70	80
	Male (%)	47.5	50	52.8	52.3	51.8	53	49.7	39.2
	Female (%)	52.5	50	47.2	47.7	48.2	47	50.3	60.8
Gender proportion of COVID-19 deaths	Age (Years)	10	20	30	40	50	60	70	80
	Male (%)	65	63.9	65.5	64.6	67.1	64.22	60.7	46.9
	Female (%)	35	36.1	34.5	35.4	32.9	35.8	39.3	53.1
Gender proportion of COVID-19 morbidity	Age (Years)	10	20	30	40	50	60	70	80
	Male (%)	0.94	3.17	4.16	3.38	3.32	3.02	2.55	2.85
	Female (%)	1.08	3.22	3.67	2.93	2.84	2.36	2.04	2.56

**Table 2 viruses-13-01145-t002:** A comparative analysis of the principal features in COVID-19 and Influenza A(H1N1).

Feature			COVID-19	Influenza A(H1N1)	References
Epidemiology and Transmission	Fecal-oral transmission		Proved	Not proved	[51,62]
	Age composition		Most patients were older than 50	Most patients were younger than 60	
	Transmission mode		Asymptomatic/ symptomatic	Symptomatic	[2]
	Reproduction number		3	1.5	[64]
	Incubation period		4.9	1.4	[63]
Treatment	Anti-viral drug		N3/ebselen/Remdesivir	Oseltamivir and zanamivir	[72,74,75]
	CP therapy		*	*	[76,77]
	Vitamin D		*	*	[69,70]
	MSC therapy		*	Not effective	[78]
Diagnosis	CRISPR-based SHERLOCK technique		*	Not develop	[79]
	qPCR		*	*	[80]
	Gut microbiome		*	*	[81]
	Lymphopenia		*	*	[82]
	CT scan		Ground-glass opacities have frequently been placed in the periphery of lower lobes	Ground-glass opacities has a central, peripheral, or random distribution	[83]
Clinical manifestation	Acute lung injury		*	*	[56]
	Cardiovascular		*	*	[50]
	Gastrointestinal	diarrhea	*	*	[84]
		nausea	*	*	
		vomiting	*	*	
Molecular biology	Receptor for virus-cell entrance		ACE2	Sialic acid receptor	[85,86]
	Genetic material		Just one positive-sense single-stranded RNA	Eight negative-sense single-stranded RNA	[87]
	Location of replication		DMV (cytoplasm)	Nucleus	[31]

* These things are ongoing as it’s a current issue and now these are usable.

**Table 3 viruses-13-01145-t003:** Vaccines candidate for the COVID-19 infection.

Vaccine Type	Vaccine	Producing Company
Inactivated vaccine	A full inactivated virus with formalin and alum adjuvant	Sinovac
Inactivated virus	Inactivated SARS-CoV-2	Beijing Institute of Biological Products, Sinopharm
Inactivated virus	Inactivated SARS-CoV-2	Wuhan Institute of Biological Products, Sinopharm
Inactivated virus	Inactivated SARS-CoV-2	Institute of Medical Biology, Chinese Academy of Medical Sciences
Subunit vaccine	S protein fusion with adjuvant and M-matrix	Novavax
Non-amplifiable viral vector vaccine	Intramuscular recombination vaccine on adenovirus type 5 (Ad5-nCoV) vector	CanSino Biological Incorporation, Beijing Institute of Biotechnology, Canadian Center for Vaccinology
Non-amplifiable viral vector vaccine	chimpanzee adenovirus-based vector (ChAdOx1) vaccine	University of Oxford, AstraZeneca
Non-amplifiable viral vector vaccine	Approach 1: Dendritic cells expressing SARS-CoV-2 minigene Approach 2: Artificial supply cells expressing SARS-CoV-2 minigene	Shenzhen Geno-Immune Medical Institute
DNA vaccine	Optimized DNA vaccine prescribed with electroporation	Inovio Pharmaceuticals
DNA vaccine	Aural DNA vaccine (bacTRL-Spike) coding SARS-CoV-2 S protein	Symvivo
RNA vaccine	mRNA vaccine for S2 region of S protein of virus enclosed by nano lipid	Moderna
RNA vaccine	mRNA vaccine with lipid nanoparticle	BioNTech, Pfizer, Fosun Pharma

## Data Availability

Not applicable.

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
