# Peer review of "Progression and Trends in Virus from Influenza A to COVID-19: An Overview of Recent Studies"

_viruses, 2021, doi:10.3390/v13061145_

Round 1
Reviewer 1 Report
The review manuscript by Daemi, et al. on summarizing the COVID19 and Influenza A outbreaks is a very good effort and I would like to congratulate the authors for their effort in packaging this review. Any information dissemination serves well for controlling these public health menaces.
A few suggestion to enhance and improve the manuscript.
- Abstract- please correct the grammar- "A decade ago, the H1N1 virus emerged in Mexico and caused an estimated 60.8 million people with 12,500 deaths globally in 19 months."
- Too ambitious a statement- "We also highlight key similarities and differences between COVID-19 and influenza A in order to identify the possible origin of COVID-19". No detailed discussion on the origins of COVID19 was made. Please either mellow down the statement or modify it.
- Page 5- "Epidemiology"-Please update the statistics to reflect the very latest figures.
- Figure 3- it would be good to put the latest numbers for COVID19 and state the ongoing nature.
- Table 1: It would be great if the authors can tabulate age stratification. It's difficult but an effort would be worthwhile- improving the impact of the manuscript.
- 4.2.2. The variable of sex is an important one in COVID19 and I'm glad the authors have included this in their review but I would recommend that authors delve a bit more on the subject and include more recent publications on this aspect. for eg. the perspective in Science" Sex differences in immune responses" by Akiko Iwasaki; Science 22 Jan 2021:Vol. 371, Issue 6527, pp. 347-348 is a great content to review and include.
- Another recommendation to enhance the review- the authors can make a table outlining all the different diagnostics and treatments and treatment regimes for both COVID and Influenza A.
Overall, I recommend this for publication once the changes are made. I don't want authors to spend a great deal of effort but these requested changes will greatly enhance the review and it's impact.
Author Response
The review manuscript by Daemi, et al. on summarizing the COVID19 and Influenza A outbreaks is a very good effort and I would like to congratulate the authors for their effort in packaging this review. Any information dissemination serves well for controlling these public health menaces.
A few suggestion to enhance and improve the manuscript.
|
No. |
Comments |
Response |
Remarks |
|
1 |
Abstract- please correct the grammar- "A decade ago, the H1N1 virus emerged in Mexico and caused an estimated 60.8 million people with 12,500 deaths globally in 19 months." |
Respected reviewer, we have corrected your point of concern. |
Comment acknowledged |
|
2 |
Too ambitious a statement- "We also highlight key similarities and differences between COVID-19 and influenza A in order to identify the possible origin of COVID-19". No detailed discussion on the origins of COVID19 was made. Please either mellow down the statement or modify it. |
Respected Reviewer, we have changed the line of concern. Moreover, we included some lines regarding origin of COVID-19 in introduction section. |
Comment acknowledged |
|
3 |
Page 5- "Epidemiology"-Please update the statistics to reflect the very latest figures. |
Respected reviewer, as this COVID-19 issue is still prevailing in different countries. So, the exact value can’t be figure out. But still we tried to modify the data. Hope the edited manuscript will cover up your suggested concern. |
Comment acknowledged |
|
4 |
Figure 3- it would be good to put the latest numbers for COVID19 and state the ongoing nature. |
We tried to put latest information but still conflict between the media reports and the numbers from county’s health department prevails. |
Comment acknowledged |
|
5 |
Table 1: It would be great if the authors can tabulate age stratification. It's difficult but an effort would be worthwhile- improving the impact of the manuscript. |
Respected reviewer, we have included a table in updated manuscript. We hope this updated version will cover up your valuable suggestion. |
Comment acknowledged |
|
6 |
4.2.2. The variable of sex is an important one in COVID19 and I'm glad the authors have included this in their review but I would recommend that authors delve a bit more on the subject and include more recent publications on this aspect. for eg. the perspective in Science" Sex differences in immune responses" by Akiko Iwasaki; Science 22 Jan 2021:Vol. 371, Issue 6527, pp. 347-348 is a great content to review and include |
Respected reviewer, we have added more data according to your suggestion in updated version of manuscript. |
Comment acknowledged |
|
7 |
Please revise the references according to the journal style. Ex. "S. Dong et al., 2007" should be "Dong et al., 2007" |
Respected reviewer, we followed the CSL website for the reference outlining. We have checked out reference styling again to make all references according to journal requirement. |
Comment acknowledged |
|
8 |
Another recommendation to enhance the review- the authors can make a table outlining all the different diagnostics and treatments and treatment regimes for both COVID and Influenza A. |
Respected reviewer, we appreciate your suggestion. We tried our best to make this review better by adding more data. Hope the edited version will cover up your all suggestions. |
|

Reviewer 2 Report
Manuscript ID: viruses-1205878
The Review “Progression and trends in virus from Influenza A to COVID-19: an overview of recent studies”, aims at discussing important clinical, epidemiological, and pathological features of COVID-19 and IV-A H1N1 to underline key similarities and differences between SARS-CoV-2 and influenza A and to attempt to identify the possible origin of COVID-19.
This is another review of SARS-CoV-2 and COVID-19. Comparing epidemiological and clinical features between COVID-19 and influenza infection the authors try to add originality to the work. However, with their data it is no possible to identify the possible origin of COVID-19.
This is my major comment:
- There is difference between SARS-CoV-2 and COVID-19. SARS-CoV-2 is the pathogens caused by COVID-19. Please, refer properly to SARS-CoV-2 o COVID-19 in the manuscript.
- With their data, authors could not attempt to identify the possible origin of COVID-19, so my suggestion is to remove this issue from aims and conclusion.
- It will be more captivating read the difference between SARS-CoV-2 and H1N1 in respect to the issue taken into account in the same section. So, I suggest the authors to re-organized the paper in order to describe, compare and clearly report (eventual) key similarities and differences in the same section between the two viruses or the two infection caused by the viruses (i.e. 2.1. Epidemiological prospective of SARS-CoV-2 and H1N1, etc…)
- A native English proof-read should be helpful
Author Response
The Review “Progression and trends in virus from Influenza A to COVID-19: an overview of recent studies”, aims at discussing important clinical, epidemiological, and pathological features of COVID-19 and IV-A H1N1 to underline key similarities and differences between SARS-CoV-2 and influenza A and to attempt to identify the possible origin of COVID-19.
This is another review of SARS-CoV-2 and COVID-19. Comparing epidemiological and clinical features between COVID-19 and influenza infection the authors try to add originality to the work. However, with their data it is no possible to identify the possible origin of COVID-19.
These are my major comments:
|
No. |
Comments |
Response |
Remarks |
|
1 |
There is difference between SARS-CoV-2 and COVID-19. SARS-CoV-2 is the pathogens caused by COVID-19. Please, refer properly to SARS-CoV-2 o COVID-19 in the manuscript. |
Respected reviewer, we have made changes in manuscript according to your suggestion. We hope that revised manuscript will cover your suggested point. |
Comment acknowledged |
|
2 |
With their data, authors could not attempt to identify the possible origin of COVID-19, so my suggestion is to remove this issue from aims and conclusion |
We have updated manuscript according to your kind suggestion. |
Comment acknowledged |
|
3 |
It will be more captivating read the difference between SARS-CoV-2 and H1N1 in respect to the issue taken into account in the same section. So, I suggest the authors to re-organized the paper in order to describe, compare and clearly report (eventual) key similarities and differences in the same section between the two viruses or the two infection caused by the viruses (i.e. 2.1. Epidemiological prospective of SARS-CoV-2 and H1N1, etc…) |
Respected reviewer, we tried our best to make this review better by adding more data to compare COVID-19 and Influenza A. Hope the edited version will cover up your all suggestions. |
Comment acknowledged |
|
4 |
A native English proof-read should be helpful. |
Corrections made according to your kind suggestion. |
Comment acknowledged |
Reviewer 3 Report
Daemi et al. provide an interesting comparative review between COVID-19 and Influenza A.
Unfortunately there are some mistakes.
Maior:
In all text "COVID-19" is used as a synonym for "SARS-CoV2", the first it have to be use for the disease, the second for the virus. Please provide.
There is not a section for vaccine, I think it could make more interesting the article.
I would be reconsidered my opinion after author's text revision.
Author Response
Daemi et al. provide an interesting comparative review between COVID-19 and Influenza A. Unfortunately there are some mistakes.
|
No. |
Comments |
Response |
Remarks |
|
1 |
In all text "COVID-19" is used as a synonym for "SARS-CoV2", the first it have to be use for the disease, the second for the virus. Please provide. |
Respected reviewer, we have made according to your suggestion. We hope the revised manuscript will cover your suggested point. |
Comment acknowledged |
|
2 |
There is not a section for vaccine, I think it could make more interesting the article. |
Respected reviewer, we appreciate your suggestion. We have included the vaccine section in manuscript. Hope the edited version will cover up your concern. |
Comment acknowledged |
Round 2
Reviewer 2 Report
I'm sorry but with their data, authors could not attempt to identify the possible origin of COVID-19 or say that COVID-19 may be originated from influenza A (H1N1), so my suggestion continue to be to remove this issue.
Reviewer 3 Report
Authors response to all my comments.